# 6-Gingerol Ameliorates Hepatic Steatosis, Inflammation and Oxidative Stress in High-Fat Diet-Fed Mice through Activating LKB1/AMPK Signaling

**DOI:** 10.3390/ijms24076285

**Published:** 2023-03-27

**Authors:** Yuzhe Liu, Dong Li, Shang Wang, Ze Peng, Qi Tan, Qifeng He, Jianwei Wang

**Affiliations:** 1Chongqing Key Laboratory of Traditional Chinese Medicine for Prevention and Cure of Metabolic Diseases, College of Traditional Chinese Medicine, Chongqing Medical University, Chongqing 400016, China; 2College of Basic Medical Sciences, Chongqing Medical University, Chongqing 400016, China

**Keywords:** non-alcoholic fatty liver disease, 6-gingerol, AMPK, LKB1/STRAD/MO25

## Abstract

6-Gingerol, one of the major pharmacologically active ingredients extracted from ginger, has been reported experimentally to exert hepatic protection in non-alcoholic fatty liver disease (NAFLD). However, the molecular mechanism remains largely elusive. RNA sequencing indicated the significant involvement of the AMPK signaling pathway in 6-gingerol-induced alleviation of NAFLD in vivo. Given the significance of the LKB1/AMPK pathway in metabolic homeostasis, this study aims to investigate its role in 6-gingerol-induced mitigation on NAFLD. Our study showed that 6-gingerol ameliorated hepatic steatosis, inflammation and oxidative stress in vivo and in vitro. Further experiment validation suggested that 6-gingerol activated an LKB1/AMPK pathway cascade in vivo and in vitro. Co-immunoprecipitation analysis demonstrated that the 6-gingerol-elicited activation of an LKB1/AMPK pathway cascade was related to the enhanced stability of the LKB1/STRAD/MO25 complex. Furthermore, radicicol, an LKB1 destabilizer, inhibited the activating effect of 6-gingerol on an LKB1/AMPK pathway cascade via destabilizing LKB1/STRAD/MO25 complex stability in vitro, thus reversing the 6-gingerol-elicited ameliorative effect. In addition, molecular docking analysis further predicated the binding pockets of LKB1 necessary for binding with 6-gingerol. In conclusion, our results indicate that 6-gingerol plays an important role in regulating the stability of the LKB1/STRAD/MO25 complex and the activation of LKB1, which might weigh heavily in the 6-gingerol alleviation of NAFLD.

## 1. Introduction

Non-alcoholic fatty liver disease (NAFLD) is considered to be the most prevalent chronic liver disease worldwide. NAFLD refers to a condition characterized by liver fat accounting for over 5% of the liver weight, in the absence of excessive alcohol consumption or other known causative factors of liver disease including viral infection, drugs and so on [1]. Recent reports shows that the overall global prevalence of NAFLD among adults is approximately 25% [2] and the continually rising prevalence exerts a huge economic burden worldwide [3]. The pathogenic spectrum of NAFLD ranges from simple hepatic steatosis to non-alcoholic steatohepatitis (NASH), which might progress into fibrosis, cirrhosis, and even to hepatocellular carcinoma, ultimately. A “two hit” mechanism is reported to drive the pathogenesis of NAFLD/NASH [4]. The first hit is hepatic steatosis, which mainly derives from hepatic de novo lipogenesis (DNL) and impaired fatty acid oxidation (FAO) [5]. The second hit included enhanced lipid peroxidation, inflammation and increased oxidative stress levels [6]. Currently, there is still no approved pharmacotherapy for NAFLD. Therefore, one possible strategy to treat NAFLD might be to alleviate hepatic lipid accumulation, inflammation and oxidative stress.

Adenosine monophosphate-activated protein kinase (AMPK), a heterotrimeric complex consisting of a catalytic subunit α and two regulatory subunits (β and γ), is a key sensor and regulator for cellular energy status [7]. Both increased intracellular AMP/ATP ratio and the upstream kinases can activate AMPK [8]. Activated AMPK phosphorylates acetyl coenzyme A carboxylase (ACC) to inactivate it, decreasing malonyl-CoA and thereby relieving the inhibition of carnitine palmitoyl transferase 1α (CPT1α), which could promote the FAO in the liver [9]. Activated AMPK can also downregulate the transcription activity and expression of sterol regulatory element binding protein 1c (SREBP1c), a principal inducer of hepatic DNL [10]. The suppression of SREBP1c decreases the transcriptional activity of ACC, fatty acid synthase (FAS) and stearoyl-coenzyme A desaturase-1 (SCD1) [11]. Thus, AMPK might be a pharmacological target for the treatment of NAFLD.

Liver kinase B1 (LKB1, also known as Stk11), one of the main upstream kinases of AMPK, was also the first to be identified. LKB1 forms a heterotrimeric complex with two regulatory proteins termed STE20-related adaptor (STRAD, an inactive pseudo-kinase) and MO25 (an armadillo-repeat scaffolding-like protein). The formation of the LKB1/STRAD/MO25 complex contributes to the cytoplasmic anchoring of LKB1, and thereby activates itself to increase its catalytic activity [12,13]. Activated LKB1 phosphorylates AMPKα at the Thr172 site to activate AMPK. The phosphorylation levels of LKB1 and AMPK were significantly decreased in high-fat diet (HFD) fed mice [14]. Previous studies have demonstrated that some compounds mitigate against NAFLD via the LKB1/AMPK pathway [15,16].

Ginger (*Zingiber officinale*), one drug-food homologous plant, has been widely used for thousands of years. 6-Gingerol (6-G, PubChem CID: 442793), one of the major pharmacologically active ingredients extracted from ginger, has been found to have pleiotropic pharmacological functions, including anti-oxidative [17], anti-inflammatory [18], anti-obesity, anti-cancer [19,20], anti-hyperglycemic and immunomodulatory [21,22] effects. Previously, 6-gingerol was reported experimentally to alleviate hepatic steatosis in HFD-fed mice [23,24], as well as to mitigate intracellular lipid accumulation and to activate AMPK in free fatty acid (FFA)-induced Hepa1–6 cells [24]. It has been reported that 6-gingerol modulates the inflammatory status and the metabolic disorders in HFD-fed rats, which is associated with a 6-gingerol-induced activating effect on hepatic AMPK [25]. However, the underlying mechanism remains elusive and it is unclear about the detailed roles of AMPK in the 6-gingerol-induced treatment of NAFLD, and whether and how LKB1 is involved in the activation of AMPK. In the present study, we intended to evaluate the efficacy of 6-gingerol against NAFLD in terms of lipid accumulation, inflammation and oxidative stress, and to investigate the mechanisms of the LKB1/AMPK pathway in 6-gingerol-induced alleviation of NAFLD.

## 2. Results

### 2.1. 6-Gingerol Improves Disorders of Glycolipid Metabolism in High-Fat Diet (HFD)-Fed Mice

During 8 weeks of HFD feeding for modeling, compared with the ND group mice, the MOD group mice showed a slightly lower average food intake (Appendix A). However, a significant increase in body weight (BW) was observed in the MOD group mice (Appendix A), which was regarded as a symbol of successful modeling. After 5 weeks of administration with 6-gingerol, HFD-induced weight gain was significantly depressed by 6-gingerol, especially a high dose 6-gingerol (Figure 1B). However, 6-gingerol had no effects on food intake in HFD-fed mice (Figure 1C), which means equal energy intake among mice from these three groups. An oral glucose tolerance test (OGTT) curve and the area under the curve (AUC) showed that the blood–glucose fluctuations were more stable in the GL and GH group mice than in mice of the HFD group (Figure 1D). In addition, we found that 6-gingerol significantly decreased fasting serum glucose and insulin levels in HFD-fed mice, as well as homeostatic model assessment for insulin resistance (HOMA-IR) (Figure 1E–G). Additionally, significant increases in serum triglyceride (TG) and total cholesterol (TC) levels were observed in HFD-fed mice, which were also markedly inhibited by 6-gingerol (Figure 1H,I). Taken together, these data show that 6-gingerol improves disorders of glycolipid metabolism in HFD-fed mice.

### 2.2. 6-Gingerol Attenuates Hepatic Steatosis and Inflammation in HFD-Fed Mice

Hematoxylin and eosin (H&E) staining showed that large amounts of steatotic vacuoles and inflammatory foci were observed in the liver of the HFD group mice, and 6-gingerol exerted an alleviative effect on this pathological state (Figure 2A). Oil Red O (ORO) staining showed that numerous varying sizes of lipid droplets were deposited in the liver of the HFD group mice. However, the volume and amount of lipid droplets were remarkably reduced by 6-gingerol (Figure 2B,C). Additionally, the liver weight and liver index (ratio of liver weight to body weight) were both increased in the HFD group mice relative to the ND group mice, which were significantly reduced by 6-gingerol, especially by high-dose 6-gingerol (Figure 2D,E). Consistent with the alleviation of lipid droplets as shown in ORO staining, 6-gingerol also significantly subdued the HFD-elevated hepatic TG and TC levels (Figure 2F,G). In addition, serum alanine transaminase (ALT) and aspartate transaminase (AST), as liver injury markers, were obviously increased in the HFD group mice liver, whereas 6-gingerol significantly inhibited the elevation of ALT and AST levels (Figure 2H,I). The HFD group mice had higher hepatic concentrations of TNF-α and IL-6 protein compared to the ND group, while the hepatic inflammatory cytokines in both the GL and GH group mice were lower than those in the HFD group mice (Figure 2J,K). Collectively, these data suggest that 6-gingerol attenuates hepatic steatosis and inflammation in HFD-fed mice.

### 2.3. 6-Gingerol Mitigates Hepatic Oxidative Stress in HFD-Fed Mice

Given that oxidative stress plays a critical role in the progression of NAFLD, we examined whether 6-gingerol affected hepatic oxidative stress induced by HFD. As expected, remarkable increases in the hepatic reactive oxygen species (ROS) level and malondialdehyde (MDA) content were observed in the HFD group mice, which were both significantly inhibited by 6-gingerol (Figure 3B,C). Additionally, the mitigative effect of 6-gingerol on HFD-induced oxidative stress was further confirmed by an immunofluorescence analysis for nitro-tyrosine (Figure 3A), a representative marker of post-translational modifications mediated by reactive nitrogen species (RNS). What is more, the wheat germ agglutinin (WGA)-marked hepatocellular membrane showed the disturbed liver structure in the HFD group mice (Figure 3A), which was reversed by 6-gingerol, indicating its therapeutic effect. Moreover, the catalytic activity of catalase (CAT) and glutathione peroxidase (GPx) were decreased in the liver of the HFD group mice, which were both significantly reversed by 6-gingerol (Figure 3D,E). Taken together, these results indicate that 6-gingerol mitigates hepatic oxidative stress in HFD-fed mice.

### 2.4. RNA-Seq and qRT-PCR Analysis Reveals the Effects of 6-Gingerol on Metabolism-Related Gene Expression in the Liver of HFD-Fed Mice

In order to systematically investigate the underlying mechanism of 6-gingerol against NAFLD, RNA-Seq was performed to analyze the effect of 6-gingerol on the hepatic mRNA transcription profile in HFD-fed mice. A Venn diagram showed that 595 genes were differently expressed among three groups due to HFD-feeding and 6-gingerol treatment. Relative to the ND group, 635 genes and 273 genes were, respectively, upregulated and downregulated in the HFD group mice liver, 398 and 197, respectively, of which were downregulated and upregulated by 6-gingerol (Figure 4A,B). The differentially expressed genes (DEGs) cluster analysis indicated that these DEGs showed significant intergroup differences (Figure 4C). As shown in Figure 4D, the KEGG pathway enrichment analysis indicates that DEGs were mainly involved in the PPAR signaling pathway, AMPK signaling pathway and metabolism-related pathway, including cholesterol metabolism, carbon metabolism, fatty acid metabolism, glycerolipid metabolism and fatty acid degradation and glutathione metabolism. Therefore, we examined the expression of genes involved in DNL and FAO. The qRT-PCR analysis results showed that relative to the ND group, DNL-related genes including *Srebf1*, *Acaca*, *Fasn* and *Scd1* were remarkably upregulated in the liver of the HFD group mice, which were significantly inhibited by 6-gingerol (Figure 4E). Additionally, 6-gingerol also reversed the HFD-induced downregulations of FAO-related genes *Cpt1α*, *Pparα* and *Pgc1α* in the liver (Figure 4F). Collectively, these data suggest the AMPK signaling pathway is highly involved in the 6-gingerol-induced treatment of NAFLD.

### 2.5. 6-Gingerol Activates LKB1/AMPK Pathway and Strengthens LKB1/STRAD/MO25 Complex Stability In Vivo

Consistent with the mRNA expression in the liver, we found that 6-gingerol strongly inhibited the protein expressions of SREBP1, ACC and FAS, and promoted the protein expressions of p-ACC and CPT1α in HFD-fed mice livers (Figure 5A,B). Then, we examined the protein expressions of p-AMPK, t-AMPK, p-LKB1 and t-LKB1. As expected, the phosphorylation levels of LKB1 and AMPK were significantly elevated by 6-gingerol in the liver of HFD-fed mice (Figure 5C,D). Considering the activation mechanism of LKB1, we evaluated the LKB1/STRAD/MO25 complex stability by co-immunoprecipitation (Co-IP) analysis. The results showed that HFD feeding reduced the interaction of LKB1 with STRAD and MO25 in the liver of the HFD group mice, indicating the destabilization of the LKB1/STRAD/MO25 complex. However, 6-gingerol regained the binding activity of LKB1 to MO25 and STRAD in HFD-fed mice liver without affecting their protein expressions, which indicated that 6-gingerol increased the stability of the LKB1/STRAD/MO25 complex in the liver of HFD-fed mice (Figure 5E,F). Taken together, these results suggest that 6-gingerol-induced activation of the LKB1/AMPK pathway cascade is related to the enhanced stability of the LKB1/STRAD/MO25 complex in the mice liver.

### 2.6. 6-Gingerol Alleviates Intracellular Accumulation of Lipid and Oxidative Stress, and Activates LKB1/AMPK Pathway in Palmitic Acid (PA)-Induced HepG2 Cells

To further ascertain whether 6-gingerol could alleviate the intracellular accumulation of lipid and oxidative stress, and to figure out how 6-gingerol effects the LKB1/AMPK pathway, HepG2 cells were exposed to palmitic acid (PA) to induce an in vitro NAFLD model. Cell Counting Kit-8 (CCK-8) results showed that PA (50, 100, 150 and 200 μM) and 6-gingerol (5, 10, 20 and 40 μM) did not significantly affect the cell viability, and the cell survival rates are all above 80% (Appendix A). Consequently, treatment with 200 μM PA for 24 h was employed in the subsequent experiments.

Nile red staining showed that 6-gingerol significantly inhibited the lipid accumulation in PA-treated HepG2 cells (Figure 6B), which was correspondingly evidenced by a flow cytometry analysis (Figure 6C). Furthermore, the PA-induced elevated intracellular TG level was significantly decreased by 6-gingerol (Figure 6D). As shown in (Figure 6A,E), PA-elevated intracellular ROS accumulation was significantly decreased by 6-gingerol. Consistent with the results in vivo, after treatment with 6-gingerol, the protein expressions of SREBP1, ACC and FAS were significantly suppressed, and the protein expressions of p-ACC and CPT1α were obviously promoted (Figure 6F). Moreover, the phosphorylation levels of LKB1, AMPK and ACC were all significantly increased by 6-gingerol (Figure 6H,I). Collectively, these data indicate that 6-gingerol alleviates the PA-induced intracellular accumulation of lipid and ROS, and activates the LKB1/AMPK pathway cascade in vitro.

### 2.7. Radicicol Reverses the Alleviation of 6-Gingerol on PA-Induced Intracellular Accumulation of Lipid and ROS via Blocking the Enhancing Effect of 6-Gingerol on LKB1/STRAD/MO25 Complex Stability

We used radicicol, an LKB1 destabilizer, to ascertain the relationship between the stability of the LKB1/STRAD/MO25 complex and the activation of the LKB1/AMPK pathway cascade in the treatment with 6-gingerol, as well as the alleviative effect on the PA-induced intracellular accumulation of lipid and ROS. CCK-8 results showed that radicicol (5 μM) did not significantly affect cell viability (Appendix A). ORO staining showed that radicicol counteracted the alleviative effect of 6-gingerol on PA-induced intracellular lipid accumulation in HepG2 cells (Figure 7B). The effect of 6-gingerol on lessening intracellular TG in steatotic HepG2 cells was also blunted in the presence of radicicol (Figure 7C). Additionally, radicicol also blocked the 6-gingerol-elicited relieving effect on intracellular ROS accumulation caused by PA in vitro (Figure 7A,D).

Concomitant with the remarkable inhibition of LKB1 and AMPK phosphorylation by radicicol (Figure 7G,H), the suppressive effect of 6-gingerol on the protein expressions of SREBP1, ACC and FAS both disappeared in the presence or absence of PA (Figure 7E,F). Moreover, in the presence of radicicol, 6-gingerol failed to promote the protein expression of CPT1α (Figure 7E,F) and did not increase the phosphorylation level of ACC (Figure 7E,H). In vitro Co-IP results showed that PA weakened the interaction of LKB1 with STRAD and MO25 in HepG2 cells, suggesting the destabilization of the LKB1/STRAD/MO25 complex. However, 6-gingerol restored the binding activity of LKB1 to STRAD and MO25 without affecting their protein expression levels, which was inhibited by radicicol, indicating that 6-gingerol increased the stability of the LKB1/STRAD/MO25 complex in the liver of HFD-fed mice (Figure 7I,J). Additionally, we detected and compared the activity level of LKB1 in PA-induced HepG2 cells treated with radicicol and those treated with 6-gingerol. As shown in Appendix A, the results showed that PA treatment lowered the phosphorylation of LKB1, and PA treatment plus radicicol further decreased the phosphorylation level of LKB1, whereas 6-gingerol reversed the further decrease in LKB1 phosphorylation induced by radicicol. Taken together, these results suggests that 6-gingerol alleviates the intracellular accumulation of lipid and ROS induced by PA via enhancing the stability of the LKB1/STRAD/MO25 complex and thus activating the LKB1/AMPK pathway cascade.

### 2.8. Molecular Docking Was Carried to Further Ascertain the Regulation Mechanisms of 6-Gingerol on LKB1

The docking calculation result, with the side chains of the residues around the binding pocket set as rigid, shows that 6-gingerol can bind inside the pocket and form hydrogen bonds with Tyr125 and Lys120 with docking energy less than or equal to −5.2 kcal/mol (Figure 8). The predictable interaction between 6-gingerol and LKB1 might be related to the binding activity of LKB1 to STRAD and MO25.

## 3. Discussion

Extensive evidence indicates that 6-gingerol exerts a protective effect on different types of hepatic steatosis models. It was reported that 6-gingerol attenuates hepatic steatosis by relieving inflammation via suppressing the NF-κB signaling pathway in HFD-fed golden hamsters [26] and in methionine and choline-deficient (MCD) diet-fed mice [27]. In recent years, 6-gingerol was reported to experimentally alleviate an inflammatory state and metabolic disorder in HFD-fed rats via the AMPK-NF-κB pathway [25]. Additionally, it has been reported that 6-gingerol ameliorates aging-related hepatic steatosis [28]. However, current anti-inflammatory mechanisms have limitedly elucidated the nature of 6-gingerol-induced alleviation of fatty liver. Our data show that treatment with 6-gingerol was able to effectively improve overall disorders of glucolipid metabolism and prevent hepatic steatosis in HFD-fed mice. Furthermore, both in vivo and in vitro results indicate that 6-gingerol could alleviate lipid accumulation, inflammation and oxidative stress, thus indicating that 6-gingerol does not rely solely on an inflammatory pathway to improve steatosis.

In order to systematically investigate the underlying mechanism, RNA-Seq was used to evaluate the effect of 6-gingerol on hepatic mRNA transcription profiles in NAFLD mice. The results indicated the key regulatory effect of the AMPK pathway in 6-gingerol-induced amelioration of NAFLD. qRT-PCR and Western blotting analysis further validated the 6-gingerol-elicited activation of the LKB1/AMPK pathway cascade in HFD-fed mice livers. Then, Co-IP analysis in vivo showed that the activation of the LKB1/AMPK pathway was connected to the stability of the LKB1/STRAD/MO25 complex regulated by 6-gingerol. Additionally, the in vitro results showed that 6-gingerol alleviated intracellular accumulations of lipid and ROS in PA-treated HepG2 cells, and activated the LKB1/AMPK pathway cascade via strengthening the LKB1/STRAD/MO25 complex stability, which were blocked by radicicol, an LKB1 destabilizer. Our study provides new evidence and mechanisms for 6-gingerol treating NAFLD, which might provide a basis for the development of 6-gingerol as a potential drug for NAFLD.

LKB1, as the most important kinase upstream of AMPK, is well-known to form a heterotrimer with pseudo-kinase STRAD and scaffolding protein MO25, and then phosphorylates itself at multiple amino acids to increase its kinase activity, then activating a series of its downstream kinase. The formations of the LKB1/STRAD/MO25 complex help to re-localize LKB1 from the nucleus to the cytoplasm [13,29,30]. It has been proposed that the heterotrimer complex is constitutively active, providing a continuous basal level of AMPK phosphorylation, and is further enhanced by conformational γ changes imposed by the binding of AMP to the subunit [31]. The phosphorylation of LKB1 at Ser431 reportedly does not affect the kinase activity itself, but is necessary for the formation of a heterotrimer complex and its translocation to the cytoplasm [12,32]. It has been reported that the destroying of the LKB1/STRAD/MO25 complex causes a decrease in AMPK phosphorylation levels and attenuates its activity [33,34]. Long-chain acyl-CoA esters were reported to inhibit the phosphorylation of AMPK at the Thr172 site by LKB1/STRAD/MO25 [35]. The phosphorylation levels of LKB1 and AMPK were significantly decreased in HFD-fed mouse [14]. Many studies have demonstrated that multiple compounds mitigate against NAFLD via the LKB1/AMPK pathway [15,16,36,37,38]. However, the effect on the stability of the LKB1/STRAD/MO25 complex by these compounds are rarely reported. In the present study, we demonstrated that 6-gingerol promoted LKB1 to bind to STRAD and MO25 to form the heterotrimer complex, which was linked to the activation of the LKB1/AMPK pathway cascade and alleviation of NAFLD in HFD-fed mice. In vitro, we further confirmed the activation of the LKB1/AMPK pathway cascade by 6-gingerol in HepG2 cells in the presence or absence of PA, and this activation depends on the effect of 6-gingerol on enhancing the LKB1/STRAD/MO25 complex stability, which was further evidenced by the treatment with radicicol.

AMPK is considered as a promising therapeutic target for NAFLD [39,40]. AMPK phosphorylation of ACC inhibits its dimerization, causing a reduction in the activity of ACC which lowers malonyl-CoA and leads to the inhibition of DNL [41,42]. Activated AMPK suppresses the transcriptional activity of Srebp1-dependent lipogenic genes in hepatocytes [10]. Inactivated ACC leads to a reduction in malonyl-CoA, which is an inhibitor of rate-limiting enzymes for CPT1α [43]. In a previous study, 6-gingerol was reported to inhibit the expression of SREBP1, FAS, ACC, and SCD1 and promote the expression of CPT1a and PPARα [25,26,44]. As expected, in the present study, 6-gingerol indeed inhibits the expression of SREBP1, FAS, ACC and SCD1, and increases the expression of CPT1α, PPARα, and PGC1α to various degrees at the level of transcription or translation in HFD-fed mice.

Oxidative stress refers to the imbalance between the production and scavenging of oxygen free radicals in vivo or in vitro, resulting in a series of oxidative products, including ROS and RNS. Excessive oxidation of overloaded fatty acids generates large amounts of oxidative metabolites in liver [45]. These radical stress products trigger the further production of lipid peroxides, resulting in chronic damage to organelles such as mitochondria [46]. Then, the degeneration and apoptosis of organelles may lead to the collapse of the whole cell, thus causing the liver injury. Moreover, oxidative stress stimulates the release of inflammatory mediators. Finally, an imbalance of the redox state and chronic inflammatory condition alter intracellular signaling collectively, thus impair energy metabolism [47,48]. Thus, oxidative stress is a crucial role in the development of NAFLD. Consistent with the previous study [44,49], the present study indicates that 6-gingerol reduces hepatic ROS and MDA contents, and restores the enzyme activity of CAT and GPx in HFD-fed mice. Moreover, we found that hepatic RNS was also significantly reduced in by 6-gingerol treatment, which was evidenced by immunofluorescence analysis for nitro-tyrosine, a representative marker of post-translational modifications mediated by RNS. It was reported that 6-gingerol alleviates intracellular ROS in HepG2 cells caused by patulin [50] and lipopolysaccharide (LPS) [51]. A previous study reported that PA could cause an increase in intracellular ROS levels [52]. However, it was not clear about the effect of 6-gingerol on PA-induced intracellular ROS. In the present study, our results showed PA treatment indeed causes an elevation in intracellular ROS in HepG2 cells, which was significantly suppressed by 6-gingerol.

Our results demonstrate that 6-gingerol increases the stability of the LKB1/STRAD/MO25 complex, consequently resulting in activating the LKB1/AMPK pathway cascade in vivo and in vitro, as shown in Figure 9. However, a detailed mechanism deserves future investigation. Then, we performed molecular docking to identify the binding sites between 6-gingerol and LKB1 protein. Based on the results of the docking analysis, we found that the potential binding of 6-gingerol and LKB1 with favorable binding energies might account for the increasing stability of the LKB1/STRAD/MO25 complex, which, however, warrants future validation. What is more, it has been reported that 6-gingerol ameliorates hepatic steatosis via the HNF4α/miR-467b-3p/GPT1 pathway [24], which provides another research strategy. Moreover, our drug efficacy mainly concentrates in the amelioration of NAFLD. However, it is not clear whether the mechanism of 6-gingerol in the present study pertains to other disease models. A study reported that Midkine, a secreted growth factor, interacts with LKB1 and STRAD to disrupt the LKB1/STRAD/MO25 complex, thereby deactivating LKB and AMPK. Midkine was reported to promote cancer cell proliferation by repressing the LKB1-AMPK axis [53]. The expression of Midkine is abnormally upregulated in various human cancers, especially in liver, lung and breast cancer [54]. Considering that many studies have reported an anti-cancer effect of 6-gingerol on various cancers [19,21,55,56], whether the similar mechanism of 6-gingerol is present in these aforementioned disease model warrants further research.

## 4. Materials and Methods

### 4.1. Chemicals

Sodium palmitate (*#*P9767) and Oil Red O (ORO, *#*O0625) were purchased from Sigma-Aldrich (Milwaukee, WI, USA). Dimethyl sulfoxide (DMSO, #D8371), fatty acid free bovine serum albumin (BSA, *#*A8850) and Nile red (*#*N8440) was obtained from Solarbio (Beijing, China). Radicicol (*#*HY-N6769) was purchased from MedChemExpress (Monmouth Junction, NJ, USA). The primary antibodies against LKB1(3047s, 1:1000), phospho-LKB1^Ser428^ (3482s, 1:1000), AMPKα (2732s, 1:1000), phospho-AMPKα^Thr172^ (2535s, 1:1000), ACC (3676s, 1:1000), phospho-ACC^Ser79^ (11818s, 1:1000), FASN (3180s, 1:1000) and β-actin (4970s, 1:1000) were purchased from Cell Signaling Technology (Beverly, MA, USA). Antibodies for CPT1α (ab128568, 1:1000), MO25 (ab51132, 1:10000) and STRAD (ab192879, 1:2000) were bought from Abcam (Cambridge, MA, USA). Antibodies against LKB1(10746-1-AP) were obtained from Proteintech (Rosemont, IL, USA). Anti-SREBP1 (sc-13551, 1:200) and anti-Nitro-tyrosine (sc-32757) were purchased from Santa Cruz Biotechnology (Santa Cruz, Dallas, Texas, USA). Alexa Fluor^®^ 488 conjugated WGA (*#*W11261) was obtained from Thermo Fisher Scientific (Waltham, MA, USA). Rabbit Control IgG (*#*AC005) and Mouse Anti-Rabbit IgG Light Chain (*#*AS061, 1:5000) were purchased from ABclonal Biotechnology (Wuhan, China).

### 4.2. Drug Preparation and Animal Procedures

6-Gingerol (*#*Z100111, CAS: 23513–14–6, HPLC ≥ 98%) was purchased from Jingzhu Biotechnology (Nanjing, China). 6-Gingerol was dissolved in DMSO for stock and was fresh diluted in 0.5 % CMC solution (*#*CC0112, Leagene, Beijing, China) with vortex and sonication until clear for animal intragastric administration.

Thirty-two six-week-old male C57BL/6J mice (weighing 20 ± 2 g) were purchased from the Laboratory Animal Center of Chongqing Medical University and kept in a specific-pathogen-free (SPF) level of the center. All animals were maintained under 12 h light/dark conditions at 22 ± 2 °C with free access to food and water. After 1 week of acclimation, the mice were randomly divided into two groups: the normal diet (ND) group (n = 8), and the model (MOD) group (n = 24). The ND group and the MOD group were fed with a normal diet and high-fat diet (D12495, Research Diets), respectively, for 8 weeks. Body weight was recorded once a week. After 8 weeks, the MOD mice group was randomly divided into three groups according to body weight (n = 6 per group): the high-fat diet (HFD) group, the HFD + 6-gingerol low-dose (GL, 6-gingerol: 1 mg/mL) group and the HFD + 6-gingerol high-dose (GH, 6-gingerol: 2 mg/mL) group. Subsequently, mice in the ND and the HFD group, fed with a normal diet and high-fat diet, respectively, were intragastrically administered with 0.5% CMC (10 mL/kg/day) for 5 weeks. Meanwhile, mice in the GL and the GH groups, fed with a high-fat diet, were intragastrically administered with 10 mg/kg/day and 20 mg/kg/day 6-gingerol, respectively. Food intake was recorded every 3 days during administration. Briefly, the animal procedures were shown in Figure 1A. All animal experiments were approved by the Animal Ethics Committee of Chongqing Medical University.

### 4.3. Oral Glucose Tolerance Test

OGTT was carried at 4 weeks after the administration of 6-gingerol. The blood samples were obtained from the tail vein and the blood glucose level was measured by a portable glucometer (Roche, Nutley, New Jersey, USA). After 12 h fasting, fasting blood glucose (FBG) levels were detected before mice were gavaged with 2 g/kg glucose. Blood glucose levels were then measured at 30, 60, 90 and 120 min after glucose gavage. The OGTT curve was drawn and AUC for each mouse was calculated using GraphPad Prism 7.0 (La Jolla, CA, USA).

### 4.4. Serum and Hepatic Biochemical Analysis

Five weeks after the administration of 6-gingerol, all mice were anesthetized with 1% pentobarbital sodium after 12 h fasting with free access to water. Sera were obtained by centrifuging the blood at 3000 rpm for 20 min after coagulation. Fasting serum TG (A110-1-1), TC (A111-1-1), GLU (A154-1-1), ALT (C009-1-1) and AST (C010-1-1) were determined using commercially available assay kits (Nanjing Jiancheng Bioengineering Institute, Nanjing, China). Fasting serum insulin, hepatic TNF-α and hepatic IL-6 were determined using an enzyme-linked immunosorbent assay (Elisa) kit (Jiubang Biotechnology Co., Ltd., Quanzhou, China). The commercial test kits for hepatic TG (E1025) and TC (E1026) were brought from Applygen Technologies Inc. (Beijing, China). All determinations were performed according to the manufacturers’ instructions. The HOMA-IRindex was calculated by the formula: [fasting serum insulin (μU/mL) × fasting serum glucose (mmol/L)]/22.5, as described by Matthews and associates [57].

### 4.5. Histological Analysis

Fresh liver samples were fixed in 4% paraformaldehyde and embedded in paraffin after routine dehydration. Subsequently, 5 μm sections were cut and used for H&E staining. H&E staining was used to observe the pathologically morphological changes of mice liver. Frozen sections were prepared with TISSUE FREEZING MEDIUM (Leica, Bensheim, Germany). Frozen sections of 8 μm were cut and used for ORO staining and immunofluorescence. ORO staining was used to observe the lipid droplets accumulated in the mice livers. Then, the slides were observed under a light microscope (Leica, Bensheim, Germany).

### 4.6. Hepatic Oxidative Stress Detection and Immunofluorescence Analysis

Immunofluorescence analysis for nitro-tyrosine was used to detect the hepatic endogenous RNS level, and WGA was used to mark the hepatocyte membranes. Briefly, liver frozen sections were permeated by 0.1% Triton X-100, followed by being blocked with 10% goat serum at RT for 1 h. Then, the sections were incubated with anti-nitro-tyrosine (1:200) and an anti-WGA (1:400) primary antibody at 4 °C overnight, followed with incubation with Dylight549-conjugated goat anti-rabbit IgG (*#*A23320, Abbkine, Wuhan, China) at RT for 1 h. The nuclei were then stained with DAPI and the slides were observed under a fluorescence microscope (Leica, Bensheim, Germany). An amount of 10% fresh liver homogenate was prepared in ice-cold phosphate-buffered saline (PBS). Subsequently, hepatic MDA (A003-1) contents, enzymatic activity of hepatic CAT (A007-1-1) and GPx (A005-1) were detected using corresponding commercial test kits (Nanjing Jiancheng Bioengineering Institute, Nanjing, China). Hepatic ROS levels were detected using a tissue ROS detection kit (*#*HR8821) obtained from Baiaolaibo Company (Beijing, China). All detections were performed according to the manufacturers’ instructions.

### 4.7. RNA-Seq Analysis

Total RNA was extracted from mice liver samples (each group containing three samples from the ND, HFD and GH group), according to the manufacturer’s instructions. After qualification and quantification, mRNA was purified by Oligo(dT)-attached magnetic beads, and then was fragmented into small pieces in a corresponding buffer. Subsequently, the cDNA was synthesized using random hexamer-primed reverse transcription. The quality control for the product was performed on the Agilent Technologies 2100 bioanalyzer. The products from the previous step were heated, denatured and circularized by the splint oligo sequence to get the final library. Finally, the library was sequenced on the BGIseq500 platform (BGI, Shenzhen, China) and yielded paired-end 50 bases reads.

The sequencing data was filtered with SOAPnuke (v1.5.2) to get clean reads, which were stored in a FASTQ format. The clean reads were mapped to the reference genome and aligned to the reference coding gene set, and then the expression level of the gene was calculated by RSEM (v1.2.12). The DEGs analysis were performed with (1) a threshold of fold change ≥ 2.0 and (2) a Q-value ≤ 0.05. Then, the DEGs were analyzed by gene clustering analysis and Kyoto Encyclopedia of Genes and Genomes (KEGG) pathway enrichment analysis.

### 4.8. qRT-PCR Analysis

Total RNA was isolated from mouse liver tissues with TRIzol reagent (Invitrogen, Carlsbad, CA, USA) and was converted to cDNA with the Evo M-MLV RT Master Mix (Accurate Biotechnology, Changsha, China). A quantitative real-time PCR (qRT-PCR) was performed on Bio-Rad CFX 96 (Bio-Rad Laboratories, Hercules, CA, USA) with SYBR^®^ Green Pro Taq HS Premix II (Accurate Biotechnology). The relative mRNA expression levels were assessed using the 2^−ΔΔCq^ method and normalized by the Actb (used as a control gene). The primers were designed from IDT DNA Technology (Coralville, IA, USA) and synthesized by Shenggong Bioengineering Co., Ltd. (Shanghai, China). The primer sequences used for the PCR are shown in Appendix A.

### 4.9. Western Blotting Analysis

Equal amounts of extracted protein were separated using SDS-PAGE and transferred onto polyvinylidene fluoride (PVDF) membranes. The membranes were blocked with 5% skim milk at RT. Then, the membranes were incubated with primary antibodies overnight at 4 °C, followed by incubation with secondary antibodies at RT. The protein bands were detected with Enhanced Chemiluminescence (ECL) reagent (Millipore, Bedford, MA, USA). The intensity was quantified by densitometry using the Image J software and used to calculate the relative protein levels normalized by β-actin.

### 4.10. Immunoprecipitation Assays

Immunoprecipitation (IP) was performed as previously described [58]. Briefly, mice liver tissues were homogenized in ice-cold IP lysis buffer (P0013, Beyotime, Shanghai, China) containing a protease inhibitor cocktail (B14001, Bimake, Houston, TX, USA) and phosphatase inhibitor (B15001, Bimake). After preclearing with Protein A/G Plus-Agarose (sc-2003, Santa Cruz), the lysate was incubated with indicated respective antibodies, overnight at 4 °C with gentle rotation, followed by incubation with Protein A/G Plus-Agarose at 4 °C for another 4 h. After washing with an IP lysis buffer three times, the immune complex was boiled with an SDS loading buffer for 10 min. Finally, immunoblotting was performed as described above, except for the second antibody (using a light chain specific antibody).

### 4.11. Cell Culture and Treatment

HepG2 cells were purchased from the China Center for Type Culture Collection (Wuhan, China) and cultured in Minimum Essential Medium (MEM) supplemented with 10% (*v*/*v*) fetal bovine serum (FBS), penicillin (100 U/mL) and streptomycin (100 mg/mL) at 37 °C in an atmosphere with 5% CO_2_. An amount of 10 mM PA/10% fatty acid free bovine serum albumin (BSA) stock solution was prepared according to the protocol as described in a previous study [59], with a slight modification. When the cell confluence reached about 60%, after 12 h serum starvation, the cells were pretreated with 20 μM or 40 μM 6-gingerol for 24 h. Subsequently, the cells were treated with 200 μM PA plus an indicated concentration 6-gingerol for another 24 h, followed by Nile red staining and the detection of intracellular ROS and TG levels. When using radicicol as an LKB1 destabilizer, the cells were treated with 40 μM 6-gingerol plus 5 μM radicicol in the presence or absence of 200 μM PA for 24 h, followed by indicated experiments.

### 4.12. Cell Viability Assay

HepG2 cells were seeded into 96-well plates, followed by treatments with different concentrations of PA (0, 50 μM, 100 μM, 150 μM, 200 μM, 250 μM and 300 μM) and 6-gingerol (0, 5 μM, 10 μM, 20 μM, 40 μM and 80 μM) for 24 h, respectively. Then, the viability was detected using CCK-8 (Dojindo, Kumamoto, Japan), according to the manufacturer’s protocol.

### 4.13. Nile Red Staining and Flow Cytometry Analysis

The cells were fixed with 4% paraformaldehyde at room temperature (RT) for 10 min, followed by incubation with 10 μM Nile red dye working solution at 37 °C for 20 min. The nuclei were stained with Hoechst 33342 and the cells were observed by a fluorescence microscope (Leica). For the quantitative analysis, cells were collected and suspended in 10 μM Nile red dye working solution at 37 °C for 20 min with gentle rotation and stained with Hoechst 33342 for 10 min. After repeated washes, centrifugation and resuspension, the fluorescence intensity was analyzed using flow cytometry (Beckman Coulter, Inc., Miami, FL, USA).

### 4.14. Determination of Intracellular Triglyceride

Intracellular TG level in HepG2 cells was determined by a commercial test kit (E1025, Applygen Technologies Inc.), according to the manufacturer’s protocol.

### 4.15. Detection of Intracellular Reactive Oxygen Species

The intracellular ROS level was detected by a probe 2′, 7′-Dichlorodihydrofluorescein diacetate (DCFH-DA, *#*HY-D0940, MedChemExpress). Cells were incubated with 10 μM DCFH-DA at 37 °C for 30 min. After washing with PBS for three times, the cells were observed by a fluorescence microscope (Leica). For the quantitative analysis, the cells were analyzed by a fluorescence microplate reader (Synergy H1, BioTek, Winooski, VT, USA).

### 4.16. Oil Red O Staining Analysis

HepG2 cells were rinsed with pre-warmed PBS and were fixed with 4% paraformaldehyde at RT for 15 min. After a rapid rinse with 60% isopropanol, the cells were stained at 37 °C for 30 min in freshly diluted ORO working solution (three parts of ORO stock solution and two parts of ddH_2_O; 0.5% ORO stock solution was prepared in isopropanol). The excess stain was removed, and the nuclei were stained with hematoxylin. Then, the cells were observed by a light microscope (Leica).

### 4.17. Molecular Docking Analysis

Molecular docking is available for predicating the interactions between small molecules and proteins. The crystal structure of LKB1 was downloaded from the Protein Data Bank (RCSB, ID: 5WXN). The structure of 6-gingerol was downloaded from PubChem (CID: 442793) in a structure data file (SDF) format and converted into a PDBQT format with ADFR suite 1.0. The binding site analysis of LKB1 and 6-gingerol was performed using Autodock Vina 1.12. The exhaustiveness of the search was set as 32, and other parameters were set as the default. The pose with the highest score (calculated binding affinity at −5.2 kcal/mol) was visualized by PyMol 2.3.

### 4.18. Statistical Analysis

The experimental results were analyzed by GraphPad Prism 7.0. The data are presented as mean ± standard error of means (SEM) unless otherwise indicated. Comparisons among groups were analyzed by a one-way ANOVA followed by Dunnett’s test. Differences among the groups at different time points or under multiple intervention conditions were analyzed by a two-way ANOVA followed by Fisher’s least significant difference (LSD) test. The cell experiment was repeated at least three times to ensure confidence in the results. *p* < 0.05 was considered statistically significant.

## 5. Conclusions

In summary, our findings indicates that 6-gingerol-alleviation of NAFLD is related to 6-gingerol-induced regulation on the stability of the LKB1/STRAD/MO25 complex and the activation of LKB1. A schematic illustration of partial mechanisms of the mitigation of NAFLD by 6-gingerol is shown in Figure 9. These findings provide new ideas into the role and the molecular mechanism of 6-gingerol in regulating NAFLD, which might provide a promising strategy for the prevention and treatment of NAFLD.

## Figures and Tables

**Figure 1 ijms-24-06285-f001:**
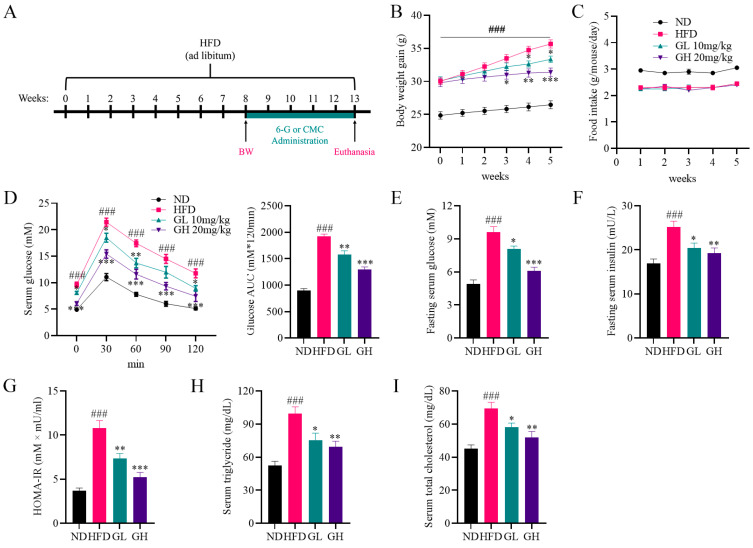
6-Gingerol improves disorders of glycolipid metabolism in high-fat diet (HFD)-fed mice. (**A**) Procedures for animal experiments. (**B**) Body weight gain and (**C**) food intake during administration. (**D**) Oral glucose tolerance test (OGTT) curve and the average area under the curve (AUC) after 4 weeks of administration with 6-gingerol. (**E**) Fasting serum glucose levels, (**F**) fasting serum insulin levels, (**G**) Homeostatic model assessment for insulin resistance (HOMA-IR), (**H**) serum triglyceride (TG) and (**I**) serum total cholesterol (TC) at the end of the experiments. Data were presented as mean ± SEM (n = 6 per group). ### *p* < 0.001 compared with the ND group; * *p* < 0.05, ** *p* < 0.01, *** *p* < 0.001 compared with the HFD group.

**Figure 2 ijms-24-06285-f002:**
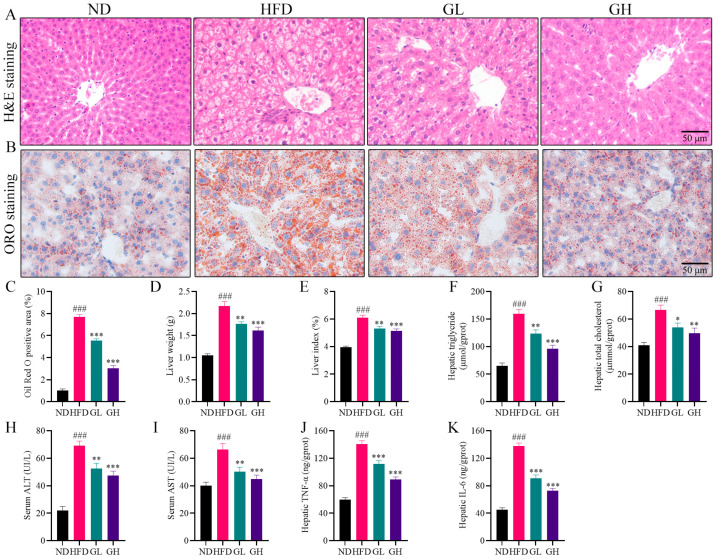
6-Gingerol attenuates hepatic steatosis and inflammation in HFD-fed mice. (**A**) Representative images of (hematoxylin and eosin) H&E-stained liver sections. (**B**) Representative images of Oil Red O(ORO)-stained liver sections. (**C**) Relative quantification of positive area of ORO staining by Image J software. (**D**) Liver weight. (**E**) Liver index (ratio of liver weight to body weight). (**F**) Hepatic triglyceride. (**G**) Hepatic total cholesterol. (**H**) Serum alanine transaminase (ALT). (**I**) Serum aspartate transaminase (AST). (**J**) Hepatic IL-6. (**K**) Hepatic TNF-α. Data were presented as mean ± SEM (n = 6 per group). ### *p* < 0.001 compared with the ND group; * *p* < 0.05, ** *p* < 0.01, *** *p* < 0.001 compared with the HFD group.

**Figure 3 ijms-24-06285-f003:**
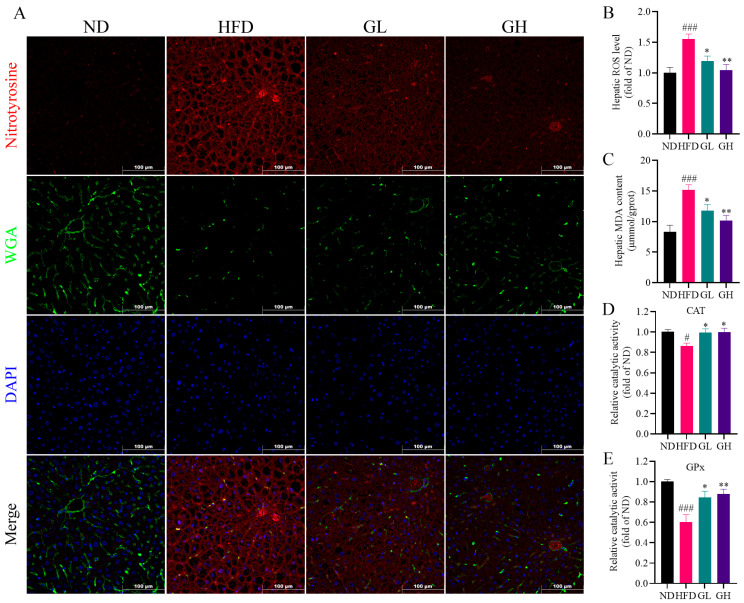
6-Gingerol mitigates hepatic oxidative stress in HFD-fed mice. (**A**) Immunofluorescence for nitro-tyrosine (red) and wheat germ agglutinin (WGA) (green) in the liver. (**B**) Hepatic reactive oxygen species (ROS) levels. (**C**) Hepatic malondialdehyde (MDA) contents. (**D**) Hepatic catalase (CAT) activity. (**E**) Hepatic glutathione peroxidase (GPx) activity. Data were presented as mean ± SEM (n = 6 per group). # *p* < 0.05, ### *p* < 0.001 compared with the ND group; * *p* < 0.05, ** *p* < 0.01 compared with the HFD group.

**Figure 4 ijms-24-06285-f004:**
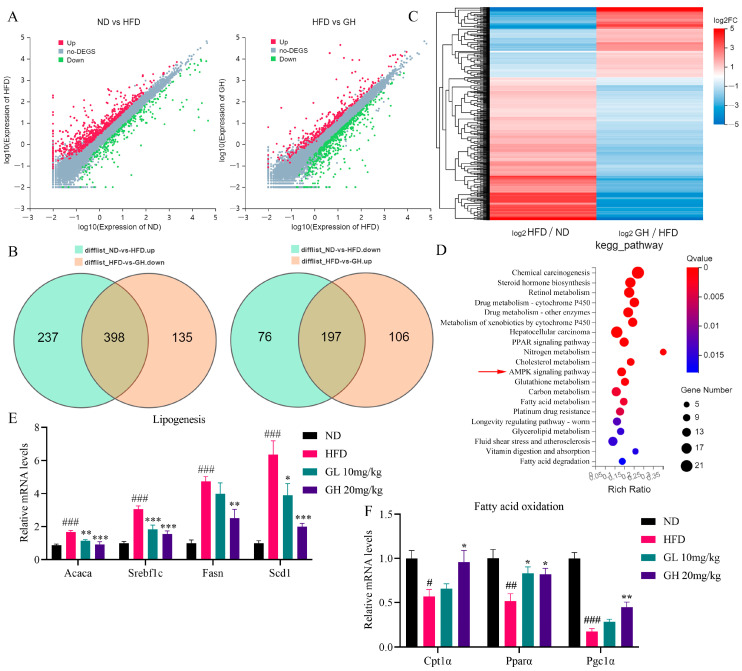
RNA-Seq and qRT-PCR analysis reveals the effects of 6-Gingerol on metabolism-related gene expression in the liver of HFD-fed mice. (**A**) Scatter plot of upregulated and downregulated differentially expressed genes (DEGs) in ND vs HFD and HFD vs GH. (**B**) Venn diagram of the overlapped DEGs between ND vs HFD and HFD vs GH. (**C**) Heat map for hierarchical cluster analysis of DEGs. (**D**) KEGG pathway enrichment analysis for DEGs. (**E**) Relative gene expressions of hepatic *Acaca*, *Srebf1*, *Fasn* and *Scd1* by qRT-PCR. (**F**) Relative gene expressions of Cpt1α, Pparα and Pgc1α by qRT-PCR. Data were presented as mean ± SEM. (n = 5/6 per group) # *p* < 0.05, ## *p* < 0.01, ### *p* < 0.001 compared with the ND group; * *p* < 0.05, ** *p* < 0.01, *** *p* < 0.001 compared with the HFD group.

**Figure 5 ijms-24-06285-f005:**
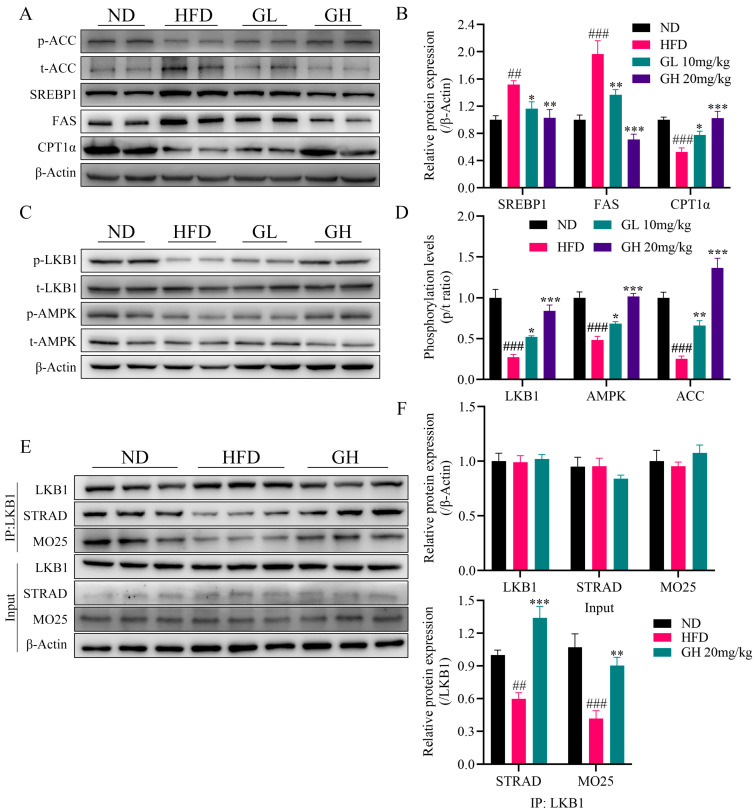
6-Gingerol activates LKB1/AMPK pathway and strengthens LKB1/STRAD/MO25 complex stability in vivo. (**A**) Protein expressions of p-ACC, t-ACC, SREBP1, FAS and CPT1α in mice livers by Western blotting. (**B**) Quantitative analysis of the relative protein expressions of SREBP1, FAS and CPT1α in liver. (**C**) Protein expressions of p-AMPK, t-AMPK, p-LKB1 and t-LKB1 in mice livers by Western blotting. (**D**) Phosphorylation levels of LKB1, AMPK and ACC in the liver. (**E**) Immunoblotting for proteins of LKB1, MO25 and STRAD before and after immunoprecipitation of LKB1 from mouse liver. (**F**) Quantitative analysis of the relative protein expressions of LKB1, STRAD and MO25. Data were presented as mean ± SEM (n = 6 per group). ## *p* < 0.01, ### *p* < 0.001 compared with the ND group; * *p* < 0.05, ** *p* < 0.01, *** *p* < 0.001 compared with the HFD group.

**Figure 6 ijms-24-06285-f006:**
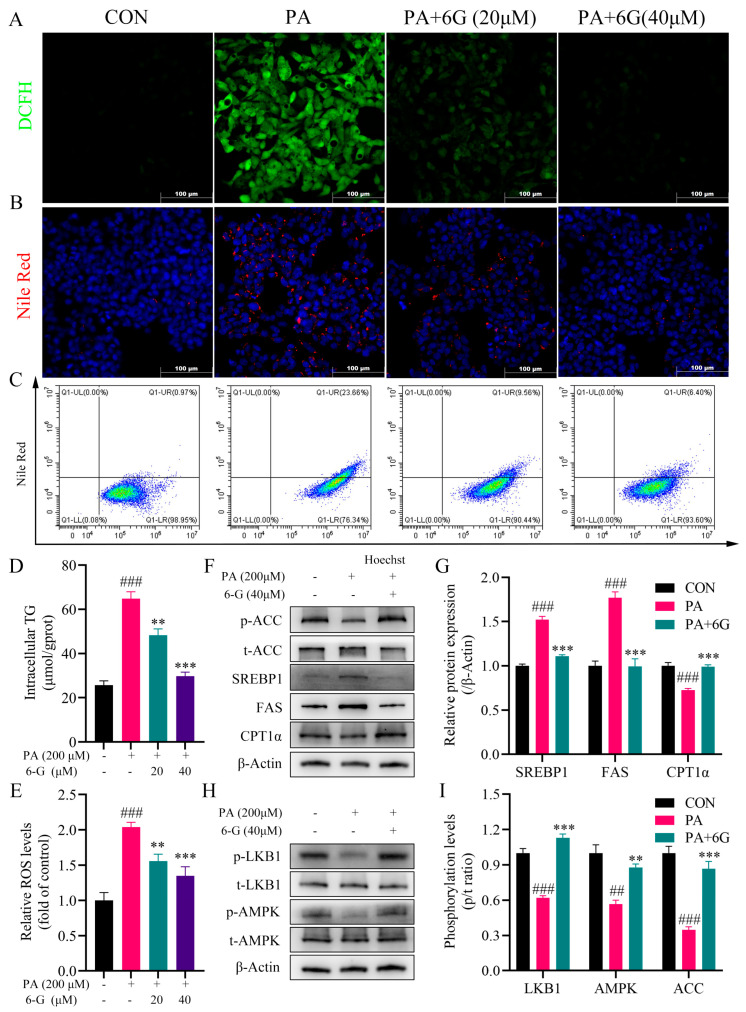
6-Gingerol alleviates intracellular accumulation of lipid and ROS, and activates LKB1/AMPK pathway in PA-treated HepG2 cells. (**A**) Representative images of immunofluorescence for intracellular ROS. (**B**) Representative images of Nile red staining. (**C**) Quantification of Nile red staining by flow cytometry analysis. (**D**) Intracellular TG level. (**E**) Intracellular ROS level. (**F**) Protein expressions of p-ACC, t-ACC, SREBP1, FAS and CPT1α in HepG2 cells by Western blotting. (**G**) Quantitative analysis of the relative protein expressions of SREBP1, FAS and CPT1α in HepG2 cells. (**H**) Protein expressions of p-AMPK, t-AMPK, p-LKB1 and t-LKB1 in HepG2 cells by Western blotting. (**I**) Phosphorylation levels of LKB1, AMPK and ACC in HepG2 cells. Data were presented as mean ± SEM. ## *p* < 0.01, ### *p* < 0.001 compared with the ND group; ** *p* < 0.01, *** *p* < 0.001 compared with the HFD group.

**Figure 7 ijms-24-06285-f007:**
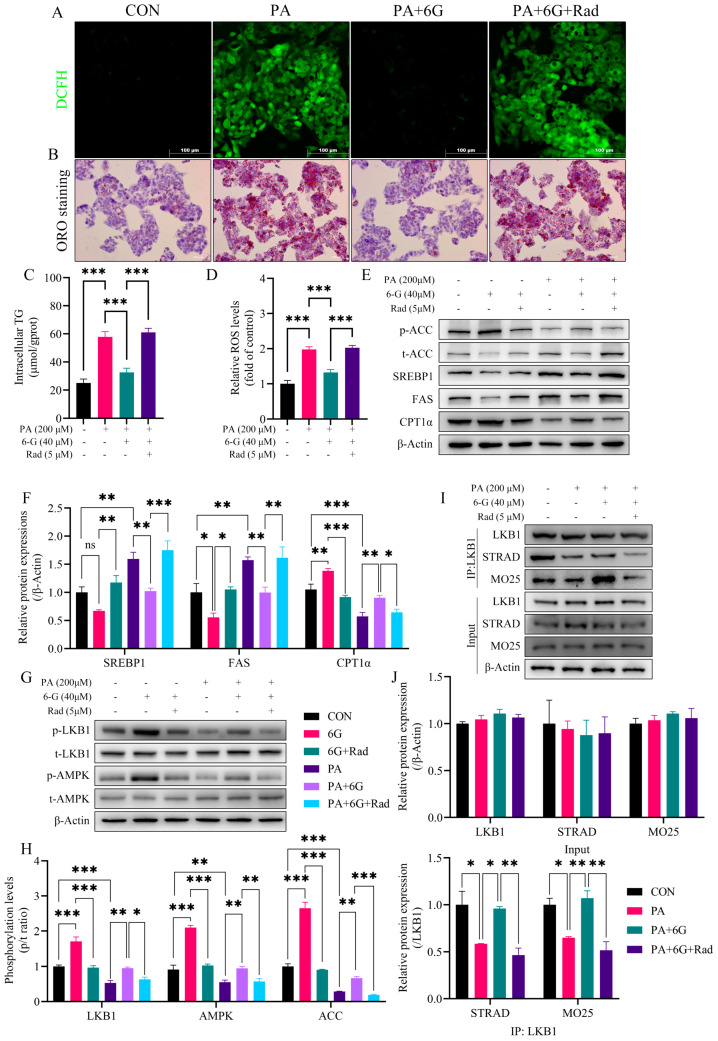
Radicicol reverses the alleviation of 6-gingerol on PA-induced intracellular accumulation of lipids and ROS via blocking the 6-gingerol-elicited enhancing effect on LKB1/STRAD/MO25 complex stability. (**A**) Representative images of immunofluorescence for intracellular ROS. (**B**) Representative images of ORO staining. (**C**) Intracellular TG level. (**D**) Intracellular ROS level. (**E**) Protein expressions of p-ACC, t-ACC, SREBP1, FAS and CPT1α in HepG2 cells by Western blotting. (**F**) Quantitative analysis of the relative protein expressions of SREBP1, FAS and CPT1α in HepG2 cells. (**G**) Protein expressions of p-AMPK, t-AMPK, p-LKB1 and t-LKB1 in HepG2 cells by Western blotting. (**H**) Phosphorylation levels of LKB1, AMPK and ACC in HepG2 cells. (**I**) Immunoblotting for LKB1, MO25 and STRAD proteins before and after immunoprecipitation of LKB1 from HepG2 cells treated as indicated. (**J**) Quantitative analysis of the relative protein expressions of LKB1, STRAD and MO25. Data were presented as mean ± SEM. * *p* < 0.05, ** *p* < 0.01, *** *p* < 0.001.

**Figure 8 ijms-24-06285-f008:**
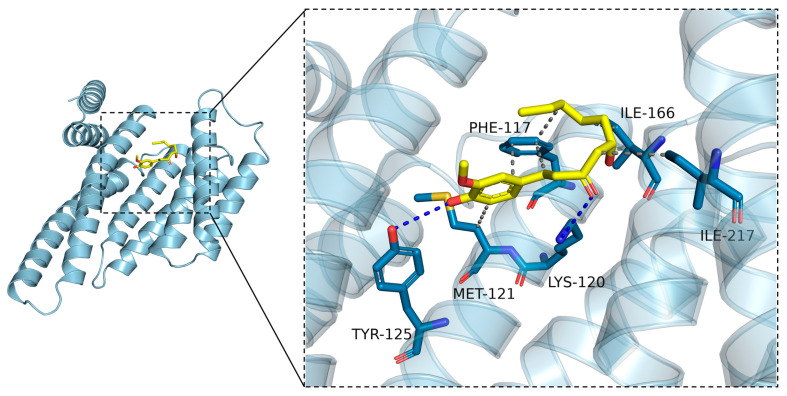
Molecular docking results for 6-gingerol and LKB1.

**Figure 9 ijms-24-06285-f009:**
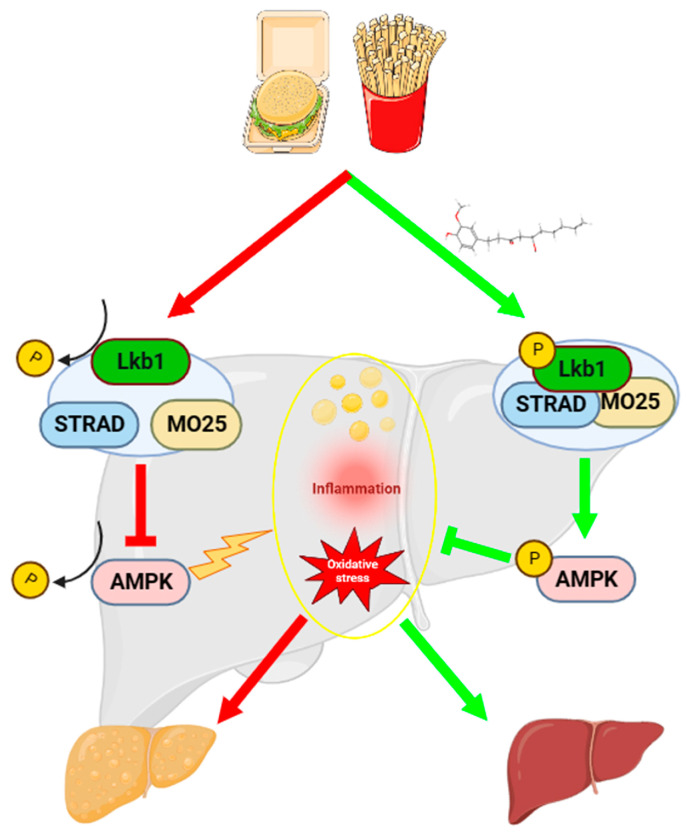
Schematic illustration showing partial mechanisms of 6-gingerol against NAFLD induced by HFD.

## Data Availability

The data that support the findings of this study are available from the first author and corresponding author upon reasonable request.

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
