# Peer review of "6-Gingerol Ameliorates Hepatic Steatosis, Inflammation and Oxidative Stress in High-Fat Diet-Fed Mice through Activating LKB1/AMPK Signaling"

_ijms, 2023, doi:10.3390/ijms24076285_

Round 1

Reviewer 1 Report

The manuscript presented to the IJSM in the section Molecular Endocrinology and Metabolism is timely and deals with a highly relevant topic. For The condition described, which is fatty liver disease, mainly of nonalcoholic origin, an unmet medical need exists. Since there are traditionally present remedies for such conditions, one of these is also ginger (lat. Zingiber officinale) with its biologically active substances, out of which the most studied and efficacious is gingerol. The experiments performed and described in this work are pertinent and deserve full attention.

State of the art in the field is very well provided, such as the experimental design. The methods used are appropriate and well-supported by the recent literature data in the field. This justifies the application of such methods in studying this silent but devastating disease by which more and more people are affected today. 

The results are clearly and comprehensively presented. The associated discussion aligns with the obtained results and is supported reasonably by the literature data. Finally, the Conclusion is bold and very ambitious, and I suggest writing it using words (e.g., potentially, promising...) to be more realistic since much more work is needed to confirm the specific mode of action of biologically active compounds of ginger.

One more suggestion is related to the way when quoting the Latin nomination of the plants. For example, in row 90, 90 Ginger (Zingiber officinale) has to be written: Ginger (lat. Zingiber officinale).

That is all from me for this review; two points to be considered:

1. Make the Conclusion less bold; keep it mild, which will make it more realistic and increase the credibility of the obtained results.  

2. Attentions should be put when using Latin expressions and nomination of the plants, these have to be distinguished from the ordinary text, and italic font has to be applied. 

I recommend accepting this publication after the spelling and language checking and changing the font of Latin expressions and names used. I would appreciate it if the editors considered my recommendation about authors being more realistic and humble when writing the Conclusion.

Author Response

Dear Reviewer and Editors,

We appreciate your constructive and helpful suggestions. Enclosed below please find our response to the reviewers’ comments and suggestions, which list the revisions we made to our manuscript. Corresponding changes are highlighted in the manuscript with “Track Changes” mode. We sincerely hope that the reviewer and the editors will be satisfied with our revisions of the manuscript.

Thank you for your consideration,

Jianwei Wang

Response to reviewer's comments and suggestions

Reviewer #1:

Comment 1: Make the Conclusion less bold; keep it mild, which will make it more realistic and increase the credibility of the obtained results.

Response: Thank you for this helpful suggestion. The Conclusion part of the abstract has been revised as follows “In conclusion, our results indicates that 6-gingerol plays an important role in regulating the stability of LKB1/STRAD/MO25 complex and the activation of LKB1, which might weigh heavily in 6-gingerol alleviation of NAFLD.” (Line 30-33). The Conclusion part of the manuscript has been revised as follows “In summary, our findings indicates that 6-gingerol-alleviation of NAFLD is related to 6-gingerol-induced regulation on the stability of LKB1/STRAD/MO25 complex and the activation of LKB1. A schematic illustration of partial mechanisms of the mitigation of NAFLD by 6-gingerol is shown in Figure 9. These findings provide new ideas into the role and the molecular mechanism of 6-gingerol in regulating NAFLD, which might provide a promising strategy for the prevention and treatment of NAFLD” (Line 583-588).

Comment 2: Attentions should be put when using Latin expressions and nomination of the plants, these have to be distinguished from the ordinary text, and italic font has to be applied.

Response: Thank you for this helpful suggestion. The Latin expressions and nomination of ginger “Zingiber officinale” has been revised to italic font “Zingiber officinale” (Line 71). Besides, 1 ) we checked the spelling of English words in the manuscript, unifying the word spellings into American spelling, including “signalling” to “signaling”, “analyse” to “analyze” and “favourable” to “favorable”. 2 ) We corrected some misspelling words, including “overlaped” To “overlapped” (Line 184).

Reviewer 2 Report

This study entitled "6-Gingerol Ameliorates Hepatic Steatosis, Inflammation and Oxidative Stress in High-Fat Diet-Fed Mice" was designed to determine potential Hepatic Steatosis, inflammation and oxidative stress effect of 6-gingerol through activation of LKB1/AMPK in vivo and in vitro. Only a few minor issues need to be addressed:

1.     The title is too simple. Please consider revising the title to include the conclusions of the study. For example, 6-Gingerol Ameliorates Hepatic steatosis, Inflammation and Oxidative Stress Effects through Activation of LKB1/AMPK signaling.

2.  The words limit for the abstract is 200 words. But the abstract of this  manuscript is 246 words. Please correct according to journal guidelines. 

      3.     Please present data on food intake during modeling (8 weeks).

      4.     Cell viability was measured for PA and 6-gingerol in vitro.  However, cell viability data for radicicol are absence. 

5.     In Fig. 6. and Fig. 7., Correct SREBF1 to SREBP1

6.     Line 394-398 : The author explains that 6-gingerol stabilizes LKB1, which is blocked by radicicol, and activates the LKB1/AMPK pathway by stabilizing the LKB1/STRAD/MO25 complex. However, in order to explain this point, it is necessary to present data on the activity level of LKB1 in PA-induced cells treated with radicicol and compare with those treated with 6-gingerol.

      7.      Add the manufacturer's country, city information when the reagent is first present. 

    8.   How many fields per image and how many liver samples were analyzed per group of animals in H&E staining?

Author Response

Dear Reviewer and Editors,

We appreciate your constructive and helpful suggestions. Enclosed below please find our response to the reviewers’ comments and suggestions, which list the revisions we made to our manuscript. Corresponding changes are highlighted in the manuscript with “Track Changes” mode. We sincerely hope that the reviewer and the editors will be satisfied with our revisions of the manuscript.

Thank you for your consideration,

Jianwei Wang

Response to reviewer's comments and suggestions

Reviewer #2:

Comment 1: The title is too simple. Please consider revising the title to include the conclusions of the study. For example, 6-Gingerol Ameliorates Hepatic steatosis, Inflammation and Oxidative Stress Effects through Activation of LKB1/AMPK signaling.

Response: Thank you for this helpful suggestion. The title has been revised to “6-Gingerol Ameliorates Hepatic Steatosis, Inflammation and Oxidative Stress in High-Fat Diet-Fed Mice through Activating LKB1/AMPK signaling” (Line 2 and 3).

Comment 2: The words limit for the abstract is 200 words. But the abstract of this manuscript is 246 words. Please correct according to journal guidelines.

Response: Thank you for this helpful suggestion. The abstract of this manuscript has been revised to include 197 words as follows: “6-Gingerol, one of the major pharmacologically active ingredients extracted from ginger, has been reported experimentally to exert hepatic protection in non-alcoholic fatty liver disease (NAFLD). However, the molecular mechanism remains largely elusive. RNA sequencing indicated the significant involvement of AMPK signaling pathway in 6-gingerol-induced alleviation of NAFLD in vivo. Given the significance of LKB1/AMPK pathway in metabolic homeostasis, this study aims to investigate its role in 6-gingerol-induced mitigation on NAFLD. Our study showed that 6-gingerol ameliorated hepatic steatosis, inflammation and oxidative stress in vivo and in vitro. Further experiment validation suggested that 6-gingerol activated LKB1/AMPK pathway cascade in vivo and in vitro. Co-immunoprecipitation analysis demonstrated that 6-gingerol-elicited activation of LKB1/AMPK pathway cascade was related to the enhanced stability of LKB1/STRAD/MO25 complex. Furthermore, radicicol, an LKB1 destabilizer, inhibited the activating effect of 6-gingerol on LKB1/AMPK pathway cascade via destabilizing LKB1/STRAD/MO25 complex stability in vitro, thus reversing 6-gingerol-elicited ameliorative effect. In addition, molecular docking analysis further predicated the binding pockets of LKB1 necessary for binding with 6-gingerol. In conclusion, our results indicates that 6-gingerol plays an important role in regulating the stability of LKB1/STRAD/MO25 complex and the activation of LKB1, which might weigh heavily in 6-gingerol alleviation of NAFLD” (Line 16-33).

Comment 3: Please present data on food intake during modeling (8 weeks).

Response: Thank you for this helpful suggestion. The data on food intake during modeling (8 weeks) has been revised into Fig. S1 in the supplement as follows:

Figure S1. (A) Food intake during modeling.

Comment 4: Cell viability was measured for PA and 6-gingerol in vitro. However, cell viability data for radicicol are absence.

Response: Thank you for this helpful suggestion. Cell viability for radicicol has been revised into Fig. S2 in the supplement as follows:

Figure S2. Cell viability under treatments with (C) 5μM Radicicol.

Comment 5: In Fig. 6. and Fig. 7., Correct SREBF1 to SREBP1

Response: Thank you for this helpful suggestion. We have corrected the protein name of “SREBF1” to “SREBP1” as shown in Fig. 5A, Fig. 6F and Fig. 7E. Accordingly, we have also unified the protein name from “SREBF1” to “SREBP1” in the manuscript.

Figure 5. (A) Protein expressions of p-ACC, t-ACC, SREBP1, FAS and CPT1α in mice liver by Western blotting.

Figure 6. (F) Protein expressions of p-ACC, t-ACC, SREBP1, FAS and CPT1α in HepG2 cells by western blotting.

Figure 7. (E) Protein expressions of p-ACC, t-ACC, SREBP1, FAS and CPT1α in HepG2 cells by western blotting.

Comment 6: Line 394-398: The author explains that 6-gingerol stabilizes LKB1, which is blocked by radicicol, and activates the LKB1/AMPK pathway by stabilizing the LKB1/STRAD/MO25 complex. However, in order to explain this point, it is necessary to present data on the activity level of LKB1 in PA-induced cells treated with radicicol and compare with those treated with 6-gingerol.

Response: Thank you for this helpful suggestion. According to your suggestion, we detected the activity level of LKB1 in PA-induced cells treated with radicicol and compare with those treated with 6-gingerol by western blotting.

As shown in the result of western blotting, PA treatment significantly lowered the phosphorylation level of LKB1. And PA treatment plus Radicicol further decreased the phosphorylation level of LKB1, which was reversed by 6-gingerol treatment.

Comment 7: Add the manufacturer's country, city information when the reagent is first present.

Response: Thank you for this helpful suggestion. The revisions about this comment were shown as follows:

“Sigma-Aldrich” has been revised to “Sigma-Aldrich (Milwaukee, WI, USA)” (Line 401 and 402).

“MedChem Express (USA)” has been revised to “MedChemExpress (Monmouth Junction, NJ, USA)” (Line 404 and 405).

“Cell Signaling Technology” has been revised to “Cell Signaling Technology (Beverly, MA, USA)” (Line 408).

“Abcam” has been revised to “Abcam (Cambridge, MA, USA)” Line (410).

“Proteintech (Rosemont, IL, United States)” has been to revised to “Proteintech (Rosemont, IL, USA)” (Line 411).

“Santa Cruz Biotechnology” has been revised to “Santa Cruz Biotechnology (Santa Cruz, CA, USA)” (Line 412 and 413).

“Thermo Fisher Scientific” has been revised to “Thermo Fisher Scientific (Waltham, MA, USA)” (Line 413 and 414).

“(#A23320, Abbkine)” has been revised to “(#A23320, Abbkine, Wuhan, China)” (Line 472).

“Baiaolaibo Company (China)” has been revised to “Baiaolaibo Company (Beijing, China)” (Line 479).

“(Invitrogen)” has been revised to “(Invitrogen, Carlsbad, CA, USA)” (Line 497 and 498).

“(Bio-Rad, USA)” has been revised to “(Bio-Rad Laboratories, Hercules, CA, USA)” (Line 500).

“(Coralville, USA)” has been revised to “(Coralville, IA, USA)” (Line 503).

“(B14001, Bimake)” has been revised to “(B14001, Bimake, Houston, TX, USA)” (Line 517).

“(Dojindo, Japan)” has been revised to “(Dojindo, Kumamoto, Japan)” (Line 540).

Comment 8: How many fields per image and how many liver samples were analyzed per group of animals in H&E staining?

Response: Thank you for this helpful comment. Eight fields per image and six liver samples were analyzed per group of animals in H&E staining.

Round 2

Reviewer 2 Report

The author has responded appropriately to the revision list. If you make a few more revised, I think improve the completeness of the manuscript.

1.      Please present the result of the supplementary data(Figure S1(A), Figure S2(C)) in the manuscript.

2.      Consider writing the presented experimental data into the manuscript to support the content of lines 394-398.

Author Response

Dear Reviewer and Editors,

We appreciate your constructive and helpful suggestions. Enclosed below please find our response to the reviewer’s comments and suggestions, which list the revisions we made to our manuscript (Line number showed in “simple mark” status). Corresponding changes are highlighted in the manuscript with “Track Changes” mode. We sincerely hope that the reviewer and the editors will be satisfied with our revisions of the manuscript.

Thank you for your consideration,

Jianwei Wang

Response to reviewer's comments and suggestions

Reviewer #2:

Comment 1: Please present the result of the supplementary data (Figure S1(A), Figure S2(C)) in the manuscript.

Response: Thank you for this helpful suggestion. Accordingly, we have revised the manuscript to present the result of the supplementary data (Figure S1(A), Figure S2(C)) in the manuscript as follows:

During 8 weeks of HFD feeding for modeling, compared with the ND group mice, the MOD group mice showed a slightly lower average food intake (Figure S1A). However, a significant increase in body weight (BW) was observed in the MOD group mice (Figure S1B), which was regarded as a symbol of successfully modeling. (Line 88-90)

CCK-8 results showed that radicicol (5μM) did not significantly affect the cell viability (Figure S2C). (Line 255 and 256)

Comment 2: Consider writing the presented experimental data into the manuscript to support the content of lines 394-398.

Response: Thank you for this helpful suggestion. We have revised the manuscript to put the presented experimental data into the manuscript as follows:

Additionally, we detected and compared the activity level of LKB1 in PA-induced HepG2 cells treated with radicicol and those treated with 6-gingerol. As shown in Figure S2D, results showed that PA treatment lowered the phosphorylation of LKB1. And PA treatment plus radicicol further decreased the phosphorylation level of LKB1, whereas 6-gingerol reversed the further decrease in LKB1 phosphorylation induced by radicicol. (Line 272-276)

Figure S2. (D) Protein expressions of p-LKB1 and t-LKB1 in HepG2 cells by western blotting.
